# Transcriptomic Analysis of Gibberellin-Mediated Flower Opening Process in Tree Peony (*Paeonia suffruticosa*)

**DOI:** 10.3390/plants14071002

**Published:** 2025-03-23

**Authors:** Bole Li, Qianqian Wang, Zefeng Qiu, Zeyun Lu, Junli Zhang, Qionghua He, Jiajun Yang, Hangyan Zhang, Xiangtao Zhu, Xia Chen

**Affiliations:** 1College of Horticultural Science, Zhejiang A&F University, Hangzhou 311300, China; bolerlee0128@gmail.com (B.L.); wqq20238103@zafu.edu.cn (Q.W.); c979923111@outlook.com (Z.Q.); wdmmtl7@163.com (Z.L.); 2College of Jiyang, Zhejiang A&F University, Zhuji 311800, China; 3Weifang Vocational College, Weifang 262737, China; 2004220019@sdwfvc.edu.cn; 4Zhuji City Garden Management Center, Zhuji 311800, China; noon0815hh@163.com (Q.H.); 15068980856@163.com (J.Y.); zhyzz10086@163.com (H.Z.)

**Keywords:** *Paeonia suffruticosa*, flower opening, hormone signaling, transcriptome sequencing

## Abstract

Gibberellin (GA_3_) plays a crucial role in regulating the flowering time of tree peony (*Paeonia suffruticosa* Andr.). However, its function on flower opening after dormancy release remains unclear, and its molecular mechanism need further study. We investigated the effects of exogenous GA_3_ treatments at 800 mg/L, 900 mg/L, and 1000 mg/L on the flowering process of five-year-old peony plants (‘Luhehong’) under greenhouse conditions. Our results showed that exogenous GA_3_ significantly accelerated the flower opening process. Specifically, flower buds treated with 800 mg/L and 900 mg/L GA_3_ bloomed after 42 and 45 days, respectively. In contrast, all flower buds treated with 1000 mg/L GA_3_ aborted, while only one flower bud in the control group bloomed after 56 days. Furthermore, analysis of endogenous hormone levels revealed that GA_3_ treatment rapidly increased endogenous GA_3_ levels, decreased ABA levels, and gradually increased IAA levels. Transcriptomic analysis of flower buds released from dormancy following GA_3_ treatment identified multiple key genes involved in the flower opening process of peony. Notably, members of the C2H2, C3H, ERF, bHLH, MYB, bZIP, NAC, and WRKY families showed significant differential expression. Moreover, several key genes involved in GA_3_, ABA, and IAA hormone signaling pathways were also differentially expressed. Our findings suggested that an appropriate concentration of exogenous GA_3_ treatment could accelerate the flower opening process in tree peony through multiple pathways, which would provide valuable insights into the molecular mechanisms underlying the gibberellin-mediated flower opening process in tree peony.

## 1. Introduction

Tree peony (*Paeonia suffruticosa* Andr.), one of the top ten traditional famous flowers in China, is known as the “King of Flowers” due to its large, fragrant blossoms and rich color variations. With its significant ornamental and economic value, tree peony has been widely cultivated and has gradually formed four main cultivar groups: Central Plains peony, Northwest peony, Southwest peony, and Jiangnan peony [1,2]. However, the short and concentrated flowering period of tree peony, typically from March to May, poses a challenge to maximizing its economic benefits. Effective flower regulation is crucial but remains a bottleneck for the industry due to imprecise control methods of tree peony [3].

The flowering process in plants encompasses three stages: floral transition, flower bud differentiation, and flower opening. Flower opening, a pivotal point in the development of ornamental traits in higher plants, has become a focal point in flower regulation research. Different flower opening mechanisms exist across plant species: flower opening in *Iris × hollandica* is primarily driven by elongation of the pedicel and ovary, accompanied by the shedding of bracts and sepals [4]. In *Rosa hybrida*, cell division and expansion contribute to petal movement, ultimately leading to anthesis [5]. In contrast, flower opening in *Tulipa × gesneriana* is predominantly regulated by changes in petal cell turgor, which are in turn modulated by temperature fluctuations [6].

Endogenous hormone regulation plays a crucial role in the flower opening process of most flowering plants, and gibberellin (GA), as a crucial plant hormone, can significantly promote blooming in abundant plants, including *Petunia*, *Gerbera jamesonii*, and *Dianthus ‘Carnation’* [7,8]. Additionally, in Arabidopsis, apart from promoting floral transition, GA also facilitates the development of petals, stamens, and anthers by antagonizing the repressive function of DELLA proteins [9]. Studies on cut roses have shown that the presence of GA_3_ and sucrose in the holding solution (vase solution) is necessary for complete petal expansion, and GA primarily regulates flower opening through synergistic interactions with ethylene signaling [10,11]. Tree peony seedlings are typically forced to flower in winter, leading to extensive research on flower regulation under forcing cultivation [12,13,14]. Exogenous GA_3_ application is widely used in tree peony forcing, with practical experience demonstrating its positive effects on bud break, flowering advancement, and increased flower diameter [15]. Autumn application of GA_3_ can delay leaf senescence in tree peony, advance budding and flowering times, and improve flowering rate [16]. While the function of GA_3_ on flower opening process remains unclear, and the specific regulatory mechanism need further study.

This study investigated the effects of exogenous gibberellin (GA_3_) on flower buds of five-year-old peony plants (‘Luhehong’) after dormancy release under greenhouse conditions with a constant temperature of 24 °C and a 14 h light/10 h dark photoperiod. The morphological changes of flower buds with different GA_3_ treatments were observed, and physiological indices including hormone levels, carbohydrate content, and protein content at different developmental stages were measured. Furthermore, transcriptome sequencing was employed to perform KEGG, GO, and KOG analyses to identify key differentially expressed genes, aiming to elucidate the physiological mechanisms underlying GA_3_ mediated regulation of flower opening in peony. Given that GA_3_ is among the more biologically active gibberellins in *Paeonia suffruticosa*, this research aims to provide a theoretical foundation for the judicious application of GA_3_ to achieve precise flower control and foster the advancement of the peony industry.

## 2. Results

### 2.1. Phenotypic Changes in Tree Peony Under Different Exogenous GA_3_ Concentrations

Phenotypic observations revealed significant morphological differences among the treatment groups during flower opening process. Four representative stages were identified: bud emergence, bud erect, small bell, and blooming (Figure 1). All three treatment groups and the control group (CK) reached bud emergence at approximately 13 days after treatment (DAT), with no significant differences in bud diameter. By 22 DAT, all treatment groups transitioned to the bud erect stage, while the CK group reached this stage at 30 DAT, again with no significant difference in bud diameter observed. At 30 DAT, Treatment 1 entered the small bell stage, followed by Treatment 2 at 33 DAT. In contrast, both Treatment 3 and CK exhibited developmental arrest with a majority of flower buds aborting and abscising. Notably, the bud diameter of Treatment 1 was significantly larger than that of Treatment 2 at this stage. By 42 DAT, Treatment 1 reached the blooming stage, followed by Treatment 2 at 45 DAT, with no significant difference in flower diameter between the two groups. Notably, both Treatment 3 and CK remained in the small bell stage at this time point (Figure 1 and Table 1).

Further analysis at 45 DAT examined various phenotypic parameters including leaf width, number of compound leaves, length of new shoots, total flower number, flowering rate, and average flower diameter. Results indicated that all three treatment groups exhibited enhanced vegetative growth compared to the CK group, as evidenced by increased leaf width, number of compound leaves, and length of new shoots. Moreover, Treatment 1 and Treatment 2 exhibited significantly greater total flower number, flowering rate, and flower diameter compared to both Treatment 3 and CK (Table 2).

Taken together, these results demonstrate that exogenous GA_3_ treatments at 800 mg/L and 900 mg/L significantly promoted flower opening in tree peony after dormancy release compared to the control. Notably, flowers in Treatment 1 reached the blooming stage approximately 3 days earlier than those in Treatment 2, indicating a more pronounced effect of the 800 mg/L treatment on accelerating flower opening. Conversely, the 1000 mg/L GA_3_ treatment had a negative impact on the flower opening process.

### 2.2. Analysis of Endogenous Hormone Content Changes

To investigate the effects of different treatments on endogenous gibberellin (GA_3_), abscisic acid (ABA), and indole-3-acetic acid (IAA) levels, hormone quantification was performed on samples collected at five time points from the three treatment and control groups. Results showed that exogenous GA_3_ treatment significantly increased endogenous GA_3_ levels by 7 DAT. In contrast, endogenous GA_3_ levels in the CK group remained relatively stable during this period. Interestingly, Treatment 1 and Treatment 2 exhibited a gradual increase in endogenous GA_3_ levels from 30 to 55 DAT, while Treatment 3 showed a gradual decline, suggesting GA_3_’s involvement in regulating the flower opening process (Figure 2A). Endogenous ABA levels exhibited a significant initial increase followed by a decline in all GA_3_-treated groups, with the timing of the peak varying depending on the treatment concentration. Treatment 1, Treatment 2, and Treatment 3 reached peak ABA levels at 55, 42, and 30 DAT, respectively. In contrast, ABA levels in the CK group remained relatively constant throughout the experiment, indicating that exogenous GA_3_ application stimulated ABA biosynthesis, with the rate of synthesis influenced by GA_3_ concentration (Figure 2B). Exogenous GA_3_ application also influenced endogenous IAA levels. All three treatment groups showed a significant increase in IAA levels from 0 to 30 DAT, peaking at 30 DAT, followed by a decline from 30 to 55 DAT. Conversely, the CK group, which did not receive exogenous GA_3_, exhibited a continuous decline in IAA levels throughout the 55-day period. These observations suggest that IAA biosynthesis is, to some extent, induced by GA_3_ (Figure 2C).

### 2.3. Analysis of Physiological Index Changes

Analysis of starch content in flower buds revealed a rapid decline in Treatment 1 before flowering (0–42 DAT), indicating continuous starch consumption until after flowering, when levels gradually increased. While Treatment 2 exhibited a similar trend to Treatment 1, the rate of starch decline was significantly slower. In contrast, Treatment 3 and CK, which did not flower, maintained relatively high starch levels throughout the experiment (Figure 2D). Soluble sugar content measurements showed increased demand during the pre-flowering (30–42 DAT) and flowering (42–55 DAT) stages. Treatment 1 exhibited a significant increase in soluble sugar content during these periods, while Treatment 2 showed a slower increase. Conversely, Treatment 3 displayed a rapid decline in soluble sugar levels, and the CK group maintained consistently low levels (Figure 2E). Analysis of soluble protein content in flower buds revealed consistently low levels in Treatment 1, suggesting high protein consumption during accelerated flower opening. Treatment 2 showed an initial increase (0–7 DAT) followed by a sharp decline, also reaching relatively low levels. In contrast, Treatment 3 and CK maintained higher levels throughout the experiment, significantly different from the other two groups (Figure 2F). These findings suggest that during flower opening, tree peony flower buds require substantial amounts of starch and soluble protein while accumulating high levels of soluble sugars. The absence or abnormal presence of carbohydrates may contribute to flower bud abortion.

### 2.4. Transcriptome Sequencing of Tree Peony Flower Buds Under Different Treatments

To identify genes involved in the flower opening process of tree peony, RNA-Seq analysis was performed on flower buds at 7 DAT following treatment with different GA_3_ concentrations. A total of 90.21 Gb of clean RNA-Seq data were generated from 12 samples, with each sample yielding 6.7 to 8.5 Gb of clean reads (Appendix A). The Q20 and Q30 values for these datasets were >97.07% and >91.89%, respectively. The GC content averaged 40.81% across the 12 datasets (Appendix A). A total of 104,888 Unigenes were identified and functionally annotated using the Nr, KEGG, KOG, and SwissProt databases. The annotation coverage rates were 44.05%, 44.20%, 23.90%, and 27.70%, respectively, with 47,018 genes successfully annotated across the four databases and 57,870 remaining unannotated (Appendix A). Comparison of DEG profiles between treatment groups and the CK group revealed 103 upregulated and 78 downregulated genes in Treatment 1, 931 upregulated and 516 downregulated genes in Treatment 2, and 498 upregulated and 1169 downregulated genes in Treatment 3 (Figure 3).

### 2.5. GO Enrichment and KEGG Pathway Analysis

Across the three pairwise comparisons, ‘cellular process’ and ‘metabolic process’ were the most significantly enriched GO terms within the biological process category (Figure 4A). Within the cellular component category, ‘cellular anatomical entity’ and ‘protein-containing complex’ were highly enriched, while ‘binding’ and ‘catalytic activity’ were significantly enriched within the molecular function category (Figure 4A). In the KOG functional classification, 30,357 Unigenes were categorized into 25 groups, with the most enriched being ‘Posttranslational modification, protein turnover, chaperones’ (O, 2792; 9.19%), ‘Signal transduction mechanisms’ (T, 2392; 7.87%), and ‘Translation, ribosomal structure and biogenesis’ (J, 1775; 5.84%). The least enriched categories were ‘Cell motility’ (N, 16; 0.05%) and ‘Extracellular structures’ (W, 59; 0.19%) (Figure 4B). KEGG pathway enrichment analysis across the three comparisons with the CK group revealed ‘Metabolic pathways’ as the most enriched pathway, followed by ‘Biosynthesis of various plant secondary metabolites’ (Figure 4C,D).

### 2.6. Analysis of DEGs in Tree Peony Under Different Treatments

To identify genes potentially involved in tree peony flower opening under different GA_3_ concentrations, DEGs exhibiting significant expression changes across all three pairwise comparisons were analyzed (Figure 5). The results revealed differential expression of numerous genes belonging to various families associated with flower development, including the C2H2, ERF, bHLH, MYB, bZIP, NAC, WRKY, and C3H families.

Within the bHLH family, expression levels of *PsUNE10*, *PsBEE2*, *PsbHLH25*, *PsbHLH62*, *PsbHLH51*, *PsbHLH90*, and *PsPIF4* were significantly upregulated across all three comparisons. Conversely, three bHLH family genes, *PsMYC2*, *PsbHLH117*, and *PsbHLH94*, were downregulated. In the ERF family, seven genes (*PsERF84*, *PsERF3*, *PsERF62*, *PsERF21*, *PsERF1-3*, *PsERF12*, and *PsERF114*) were significantly upregulated, while four genes (*PsDREB2A*, *PsERF109*, *PsDREB1D*, and *PsERF16*) were downregulated across all comparisons. Interestingly, *PsERF61* was upregulated in both C1 and C2 comparisons but downregulated in the C3 comparison. Eight NAC family members (*PsNAC35*, *PsNAC71*, *PsNAC86*, *PsNAC73*, *PsNAC45*, *PsNAC67*, *PsNAC90*, and *PsNAC30*) were significantly upregulated across all three comparisons, while five members (*PsJA2L*, *PsNAC47*, *PsNAC79*, *PsNAC100*, and *PsNAC98*) were downregulated. Six MYB family genes (*PsMYB4*, *PsMYB2*, *PsMYB61*, *PsRL3*, *PsSRM1*, and *PsMYB16*) were significantly upregulated, while four genes from the same family (*PsMYB102*, *PsMYB63*, *PsCSA*, and *PsMYB93*) were downregulated. Within the WRKY gene family, only *PsWRKY40* showed a significant upregulation trend across all comparisons, whereas four other genes (*PsWRKY22*, *PsWRKY29*, *PsWRKY6*, and *PsWRKY56*) were significantly downregulated. Only two C3H family genes exhibited significant expression differences: *PsZFS1* was significantly upregulated, while *PsC3H20* was significantly downregulated. Among the bZIP family members, two genes (*PsbZIP34* and *PsRF2b*) were significantly upregulated, while three (*PsCYS-3*, *PsCPC-1*, and *PsMBZ1*) were significantly downregulated across the three pairwise comparisons. Additionally, two C2H2 family genes (*PsZAT9* and *PsMGP*) were significantly upregulated, while two others (*PsZAT10* and *PsZAT11*) were significantly downregulated (Figure 5).

### 2.7. Verification of Relative Gene Expression During Flower Opening Process

Twelve putative bHLH, MYB, ERF, NAC, and petal-expansion-related genes were selected from the DEGs: *PsbHLH25*, *PsbHLH51*, *PsbHLH62*, *PsMYB02*, *PsMYB61*, *PsERF12*, *PsERF62*, *PsNAC35*, *PsNAC45*, *PsEXP10*, *PsEXP11*, and *PsXTH10*. The expression patterns of these 12 genes under CK and 800 mg/L treatment at 7 DAT were verified using qRT-PCR (Figure 6) and their expression trends were found to be similar to those obtained by RNA-seq, suggesting that the RNA-seq data reliably reflect the gene expression trends. According to the RNA-seq and qRT-PCR results, compared to CK, the expression levels of 12 genes were increased at 7 DAT, suggesting that the high levels of those genes induce the flower opening process of tree peony.

### 2.8. Analysis of DEGs in Plant Hormone Signal Transduction Pathways Under Different Treatments

Gene expression analysis revealed that the rapid increase in endogenous GA_3_ levels within 7 DAT triggered a plant hormone balance mechanism within the GA signaling pathway. Specifically, one GA biosynthesis gene, *PsGA20ox*, was significantly downregulated, while two GA biosynthesis inhibitor genes, *PsGA2ox*, were significantly upregulated in all three treatment groups. Furthermore, due to the inhibition of GA biosynthesis, the expression of two downstream GA receptor genes, *GIBBERELLIN INSENSITIVE DWARF1* (*PsGID1*), was also significantly downregulated. The decreased *PsGID1* expression led to reduced GA protein degradation signaling, resulting in the gradual accumulation of DELLA proteins and subsequent downregulation of downstream flowering promoting factors LEAFY (*PsLFY*) and *SUPPRESSOR OF OVEREXPRESSION OF CONSTANS 1* (*PsSOC1*) (Figure 7). Interestingly, the endogenous GA level balance mechanism in Treatment 1 and Treatment 2, which exhibited an early flowering phenotype, ceased by 30 DAT, and the GA synthesis pathway was reactivated. This reactivation led to a significant increase in endogenous GA_3_ levels from 30 to 42 DAT, coinciding with the flower opening process in tree peony (Figure 2A).

Within the ABA signaling pathway, five ABA receptor *PYR1-LIKE* (*PsPYL*) genes were significantly downregulated in all three treatment groups, consistent with the observed trend of ABA accumulation from 0 to 7 DAT. ABA binding to its receptor, *PsPYL*, typically inhibits the downstream phosphatase *PROTEIN PHOSPHATASE 2C* (*PsPP2C*), thereby activating downstream signaling. However, the significant downregulation of *PsPYL* in this study led to increased *PsPP2C* expression and subsequent downregulation of key positive regulators *ABA INSENSITIVE* (*PsABI*) and *ABA-RESPONSIVE ELEMENT BINDING FACTOR* (*PsABF*), ultimately alleviating the inhibitory effect on flowering through the ABA signaling pathway (Appendix A).

In the IAA signaling pathway, the cell proliferation and expansion regulator *JAGGED* (*PsJAG*) regulates two interacting factors involved in petal formation: *RABBIT EARS* (*PsRBE*) and *PETAL LOSS* (*PsPTL*). These factors positively regulate the auxin influx carrier *AUXIN RESISTANT 1* (*PsAUX1*) and the complex carrier-binding factor *AUXIN RESISTANT 4* (*PsAXR4*), promoting IAA accumulation in tissues through the downstream auxin efflux carrier *PIN-FORMED* (*PsPIN*) and its activating kinase *PINOID* (*PsPID*), ultimately promoting petal development (Appendix A). Consistent with the observed upregulation of these pathway genes, a substantial accumulation of endogenous IAA was detected in the flower buds of Treatment 1, Treatment 2, and Treatment 3 from 7 to 30 DAT, coinciding with the period of petal formation from bud emergence to the small bell stage. These findings suggest that exogenous GA_3_ treatment promotes IAA accumulation, facilitating petal formation. In contrast, the CK group, which did not receive exogenous GA_3_, exhibited a decline in IAA levels and a significantly delayed transition to the small bell stage compared to the treatment groups (Figure 1 and Figure 2C).

### 2.9. Analysis of Gene Expression Related to GA Pathway

Six putative genes related to GA pathway were selected from the DEGs: *PsGA2ox-1*, *PsGA2ox-8*, *PsGA20ox*, *PsGID1-A*, *PsGID1-B*, and *PsGID2*. The expression patterns of these six genes under CK and 800 mg/L treatment at 7 DAT were verified using qRT-PCR and their expression trends were found to be similar to those obtained by RNA-seq, suggesting that the RNA-seq data reliably reflect the gene expression trends (Figure 8). According to the RNA-seq and qRT-PCR results, compared to CK, the expression levels of *PsGA2ox-1* and *PsGA2ox-8* were increased at 7 DAT. Meanwhile, *PsGA20ox*, *PsGID1-A*, *PsGID1-B*, and *PsGID2* were decreased at 7 DAT, indicating that the genes related to GA pathway were sensitive to exogenous GA_3_ treatment.

## 3. Discussion

Flower opening, a critical event in the plant life cycle, is regulated by a complex network of factors, with plant hormone signaling pathways (GA, ABA, and IAA) playing a crucial role [17,18,19]. While forcing the cultivation of introduced tree peony remains relatively unexplored, a deeper understanding of its flowering mechanisms is essential [12,13,14]. Investigating the regulatory mechanisms governing the flower opening process in tree peony is therefore crucial for optimizing production and cultivation practices.

In lily, exogenous GA_3_ treatment has been shown to accelerate plant development and prevent bud abortion [20]. In Paphiopedilum, lateral buds often fail to develop, but exogenous GA_3_ application promotes lateral flower differentiation and blooming, accompanied by increased IAA content and upregulation of IAA transporter genes [21]. While exogenous GA_3_ generally promotes flowering, excessively high concentrations can lead to GA-dependent abortion. For instance, in blackberry, the bud break rate increased linearly with exogenous GA_3_ concentration, but the yield exhibited an inverse exponential relationship, highlighting the occurrence of GA-dependent abortion [22]. In tree peony, exogenous GA_3_ treatment significantly reduced flower bud abortion rate, with a negative correlation observed between aborted buds, sugar accumulation, and ABA content. High ABA and sugar levels were directly linked to bud abortion [23]. Consistent with previous findings, our study demonstrated that exogenous GA_3_ treatment significantly accelerated flower opening in tree peony and reduced bud abortion rate (Figure 1 and Table 1). However, an excessively high concentration (1000 mg/L) resulted in bud abortion. Hormone and physiological analyses indicated that, similar to previous reports, high ABA and sugar content contributed to bud abortion in our study (Figure 2). Interestingly, despite the absence of high ABA and sugar levels in the treatment groups, bud abortion still occurred, suggesting the involvement of other factors warranting further investigation.

In the current study, significant enrichment of multiple pathways was observed, with distinct patterns across contrasts (Figure 4C–E). Pentose and glucuronate interconversions: This pathway was prominently enriched, suggesting its critical role in cell wall modification and structural changes necessary for flower opening [24]. The upregulation of genes involved in this pathway could facilitate the softening of floral tissues, promoting petal expansion. Starch and sucrose metabolism: The enrichment of this pathway highlights the importance of energy supply and carbohydrate transport in flower development. Sugars are known to act as signaling molecules influencing floral transition and petal elongation [25]. Cutin, suberine, and wax biosynthesis: These pathways are essential for the formation of protective barriers, ensuring proper water retention and defense against pathogens during flower opening [26]. Regulation of this pathway could be critical for maintaining floral organ integrity. Plant hormone signal transduction: The presence of GA_3_, ABA, and IAA-related genes in this pathway underscores the hormonal control of flower opening. GA_3_ is known to promote petal growth, while ABA modulates stress responses, and IAA is integral to cell elongation [27]. Biosynthesis of secondary metabolites: The significant enrichment of this pathway indicates its role in producing compounds that attract pollinators or protect against environmental stressors. Flavonoid and phenylpropanoid biosynthesis were notably active, suggesting their involvement in coloration and scent production [28,29].

Numerous genes participate in the molecular mechanisms underlying flower opening. Our transcriptome data analysis revealed the involvement of gene families such as C2H2, ERF, bHLH, MYB, bZIP, NAC, WRKY, and C3H in this process (Figure 5). The C2H2 family, one of the largest zinc finger transcription factor families in plants, has been implicated in flower development. For instance, in Aquilegia, C2H2 zinc finger transcription factors promote flower development by enhancing petal cell proliferation [30]. The ERF family, another large plant transcription factor family, also plays a role in flower opening. In pineapple, 25 family members were significantly upregulated during this process [31], while in chrysanthemum, the *CmERF110* gene regulates flowering time and confers an early flowering phenotype by influencing the photoperiod [32]. Members of the bHLH transcription factor family are recognized as essential regulators of plant growth and development. In Arabidopsis, seven bHLH transcription factors (*AtFBH1*, *AtFBH2*, *AtFBH3*, *AtFBH4*, *AtMYC2*, *AtMYC3*, and *AtMYC4*) were identified as positive regulators of early flowering [33,34]. Similarly, in tree peony, bHLH transcription factor family member *PsMYCs* can accelerate flower opening [35]. The MYB transcription factor family contributes to various aspects of plant growth and development. In wheat, *TaMYB72* promotes early flowering by upregulating the flowering genes *TaHd3a* and *TaRFT1* [36]. In tree peony, *PsMYB44* accelerates anthocyanin accumulation in petals, promoting flower coloration [37]. In soybean, the bZIP family member *GmFDL19* interacts with *GmFTs* to promote early flowering [38]. NAC transcription factor family members such as *CUC1*-*CUC3*, *NAM*, and *NH16* can suppress vegetative growth while promoting floral organ formation [39,40]. In Arabidopsis, the WRKY family member *AtWRKY71* promotes flowering [41], while in wintersweet (*Chimonanthus praecox*), C3H zinc finger protein genes *CpCZF1* and *CpCZF2* participate in stamen development [42]. Our study found differential expression of members from these families in tree peony flower buds, supporting their potential involvement in the flower opening process.

Hormone signal transduction plays a vital role in regulating plant growth and development, including responses to environmental stresses and internal metabolic coordination. Among these hormones, GA signaling pathway is crucial for flowering. In this study, we quantified the endogenous levels of GA and analyzed genes involved in their respective signaling pathways using transcriptome data. Exogenous GA_3_ application led to a rapid increase in endogenous GA_3_ levels, peaking at 7 DAT (Figure 2A). *PsGA2ox*, a key enzyme in GA catabolism, is positively regulated by exogenous GA, while *PsGA20ox*, a rate-limiting enzyme in GA biosynthesis, is negatively regulated by endogenous GA [43]. Based on endogenous GA levels, exogenous GA_3_ treatment conditions, and expression patterns of GA biosynthesis genes, we hypothesize that exogenous GA_3_ application triggered a dynamic GA balance mechanism in flower buds, influencing downstream components of the GA signaling pathway. GA binds to its cytoplasmic receptor GID1, triggering conformational changes that facilitate DELLA protein binding [44,45,46], forming a GA-GID1-DELLA ternary complex [47,48]. SLY1 and GID2, F-box proteins within the SCF E3 ubiquitin ligase complex, recognize and bind the GA-GID1-DELLA complex, promoting DELLA protein ubiquitination and degradation [49]. DELLA proteins repress downstream flowering promoting factors LFY and SOC1 [50,51], while SOC1 can promote LFY expression [52,53], collectively promoting flowering. Our findings in tree peony align with the previously reported GA signaling pathway (Figure 6); although related genes were identified, their expression patterns differed. This difference in expression patterns may be due to different sequencing time points: previous studies often analyzed samples at 3 DAT, whereas we sequenced samples at 7 DAT.

## 4. Materials and Methods

### 4.1. Plant Materials and Treatment Methods

Tree peony cultivar ‘Luhehong’ (*Paeonia suffruticosa* Andr. ‘Luhehong’) used in this study was grown at the experimental field of Jiyang College, Zhejiang A&F University (Zhuji, Zhejiang, China, 29°75′52′′ N, 120°26′12′′ E). After dormancy release, the plants were transferred to a greenhouse at the same location in December 2022 and cultivated under conditions of 14 h light/10 h dark and a constant temperature of 24 °C. A total of 56 plants with similar growth status and 7–12 flower buds per plant were selected and randomly divided into four groups. GA_3_ treatments were applied daily from 29 to 31 December 2022 at 2 p.m. The control group (CK) received topical applications of ultrapure water (ddH_2_O) to the flower buds. Treatment groups received topical applications of 800 mg/L (Treatment 1), 900 mg/L (Treatment 2), or 1000 mg/L (Treatment 3) GA_3_ solution.

Flower bud samples (*n* = 3 per group) and mature functional leaves (3–6 per group) located near the terminal buds were collected at five time points: 0, 7, 30, 42, and 55 days after the initial treatment (29 December 2022, designated as day 0). Following photography, samples were immediately frozen in liquid nitrogen and stored at −80 °C until further analysis.

All materials used for physiological and biochemical assays and transcriptome sequencing were flower buds (three biological replicates per group), while all leaf materials were used solely for phenotypic measurements. Additionally, separate flower bud samples (three biological replicates) were used for quantitative real-time PCR (qRT-PCR) experiments.

### 4.2. Morphological Measurements

The initial number of flower buds per plant was recorded on day 0. After 45 days of treatment, the following phenotypic parameters were measured: number of flowers, flowering rate, flower diameter (the longest distance across the flower when viewed from above), leaf width (the widest distance across the leaf blade), number of compound leaves, and length of new shoots (distance from the terminal bud to the first internode).

### 4.3. Physiological Measurements

Endogenous hormone quantification: Twelve samples per treatment group (CK, Treatment 1, Treatment 2, Treatment 3) were analyzed at each time point (three biological replicates). The levels of three endogenous hormones, gibberellin (GA_3_) (F5052-B), indole-3-acetic acid (IAA) (F4994-B), and abscisic acid (ABA) (F50089-B), were quantified using enzyme-linked immunosorbent assay (ELISA) kits (FANKEW, Shanghai, China) at five time points: 0, 7, 30, 42, and 55 days after treatment.

Soluble protein (KMSP-2-W), soluble sugar (KT-2-Y), and starch content (DF-2-Y) determination: Similar to hormone analysis, twelve samples per treatment group were analyzed at five time points. Soluble protein content was determined using the Coomassie Brilliant Blue G-250 staining method [54], while soluble sugar and starch content were measured using the anthrone colorimetric method [55,56]. Commercially available kits (COMIN, Suzhou, China) were used for all three assays.

### 4.4. RNA Extraction, Library Construction and Sequencing

RNA was extracted from a total of 12 samples for Illumina transcriptome sequencing, including three biological replicates each of CK, Treatment 1, Treatment 2, and Treatment 3 groups. These samples were all collected at day 7. Total RNA was extracted using Trizol reagent kit (15596026CN) (Invitrogen, Carlsbad, CA, USA) according to the manufacturer’s protocol. RNA quality was assessed on an Agilent 2100 Bioanalyzer (Agilent Technologies, Palo Alto, CA, USA) and checked using RNase free agarose gel electrophoresis. After total RNA was extracted, eukaryotic mRNA was enriched by Oligo(dT) beads, while prokaryotic mRNA was enriched by removing rRNA by Ribo-Zero^TM^ Magnetic Kit (Epicentre, Madison, WI, USA). Then, the enriched mRNA was fragmented into short fragments using fragmentation buffer and reverse transcribed into cDNA with random primers. Second-strand cDNA was synthesized by DNA polymerase I, RNase H, dNTP and buffer. Then, the cDNA fragments were purified with QiaQuick PCR extraction kit (Qiagen, Venlo, The Netherlands), end repaired, A base added, and ligated to Illumina sequencing adapters. The ligation products were size selected by agarose gel electrophoresis, PCR amplified, and sequenced using Illumina novaseq 6000 by Gene Denovo Biotechnology Co. (Guangzhou, China).

### 4.5. Analysis of Differentially Expressed Genes (DEGs) and Enrichment

RNA differential expression analysis was performed by DESeq2 [57] software (the version number of R studio is 4.3.2) between two different groups (and by edgeR [58] between two samples). The genes with the parameter of false discovery rate (FDR) below 0.05, *p* below 0.05 and absolute fold change ≥ 2 were considered differentially expressed genes.

Gene Ontology (GO) [59] enrichment analysis provides all GO terms that significantly enriched in unigenes comparing to the genome background, and filter the unigenes that correspond to biological functions. Firstly, all unigenes were mapped to GO terms in the Gene Ontology database (http://www.geneontology.org/), gene numbers were calculated for every term, significantly enriched GO terms in unigenes comparing to the genome background were defined by hypergeometric test. The calculated *p*-value was gone through FDR Correction, taking FDR ≤ 0.05 as a threshold. GO terms meeting this condition were defined as significantly enriched GO terms in unigenes. This analysis was able to recognize the main biological functions that unigenes exercise.

Pathway enrichment analysis, based on all KEGG-annotated genes, identified significant enrichment of metabolic and signal transduction pathways in the DEGs relative to the whole-genome background [60]. The calculated *p*-value was gone through FDR Correction, taking FDR ≤ 0.05 as a threshold. Pathways meeting this condition were defined as significantly enriched pathways in DEGs.

### 4.6. RNA Extraction for Quantitative Real-Time PCR Analysis

Total RNA was isolated using the RNAprep Pure Plant Kit (DP441) (TianGen, Beijing, China), and its quality was evaluated using a nucleic acid analyzer (Implen Company in Germany). First-strand cDNA synthesis was performed using the PrimeScript^TM^ RT reagent Kit (RR037A) (TaKaRa, Dalian, China).

The related gene sequences were identified from the genomic data of tree peony downloaded from the genome website (https://ftp.cngb.org/pub/CNSA/data5/CNP0003098/CNS0560369/CNA0050666) (accessed on 12 September 2023) through the BlastX annotation. Primer Premier 5 software was utilized for the design of qRT-PCR primers, with *PsACT* serving as the reference gene. The expression levels of the relevant genes were assessed through quantitative real-time polymerase chain reaction (qRT-PCR) using the Light Cycler 480II Real Time PCR system (Roche, Basel, Switzerland). The reaction system consisted of 10 μL SYBR Premix Ex Taq, 2 μL cDNA, 0.8 μL each of upstream and downstream primers (10 μmol/L), and ddH_2_O to a final volume of 20 μL. The reaction procedure was 95 °C for 30 s, 95 °C for 5 s, 60 °C for 30 s, a total of 40 cycles; 95 °C for 5 s, 60 °C for 1min, 95 °C for 15 s. All qRT-PCR experiments were conducted with three biological replicates (distinct from the samples used for transcriptomics). The relative expressions were calculated by 2^−ΔΔCT^ method [35].

### 4.7. Statistical Analysis

The statistical analyses were performed using SPSS (16.0 version; SPSS Inc., Chicago, IL, USA). Data was expressed as the mean ± SE from three independent experiments with three biological replicates for each. According to Duncan’s multiple range test, differences were assessed by one-way analysis of variance (ANOVA) at *p* < 0.05.

## 5. Conclusions

Our results showed that application of 800 mg/L exogenous GA_3_ to flower buds released from dormancy significantly accelerated the flowering process and reduced bud abortion rates. These findings were supported by phenotypic observations and endogenous hormone measurements, confirming the crucial role of GA in promoting flower opening. Furthermore, transcriptome sequencing analysis revealed significant enrichment of pathways such as Plant hormone signal transduction, Starch and sucrose metabolism, and Biosynthesis of secondary metabolites. Eight gene families associated with flowering showed significant differential expression, indicating their involvement in regulating the flowering process. We focused on analyzing key genes within the GA hormone signal transduction pathway and found a strong correlation with the flower opening process. These findings provide a foundation for understanding the molecular mechanisms underlying hormone-mediated regulation of flower opening in peony.

## Figures and Tables

**Figure 1 plants-14-01002-f001:**
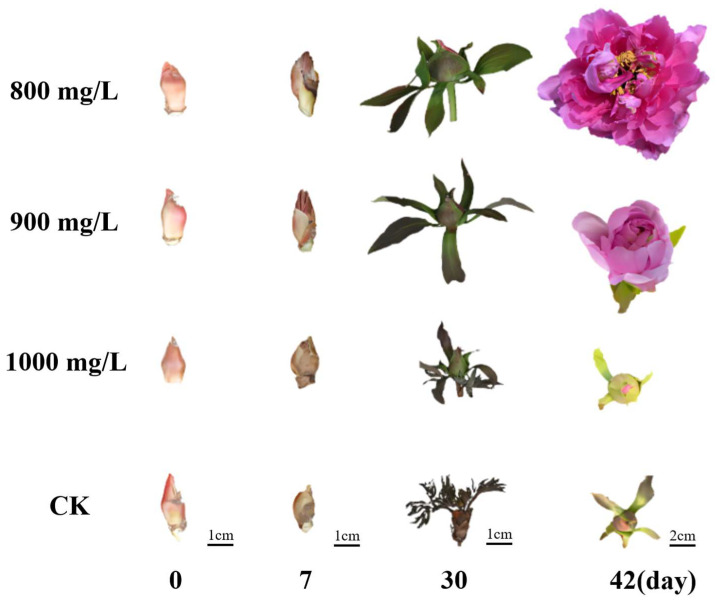
The function of exogenous hormone in flower opening of tree peony. CK (Control): the control without exogenous hormone treatment. Treatment 1 (800 mg/L GA_3_): 800 mg/L exogenous GA_3_ treatment. Treatment 2 (900 mg/L GA_3_): 900 mg/L exogenous GA_3_ treatment. Treatment 3 (1000 mg/L GA_3_): 1000 mg/L exogenous GA_3_ treatment.

**Figure 2 plants-14-01002-f002:**
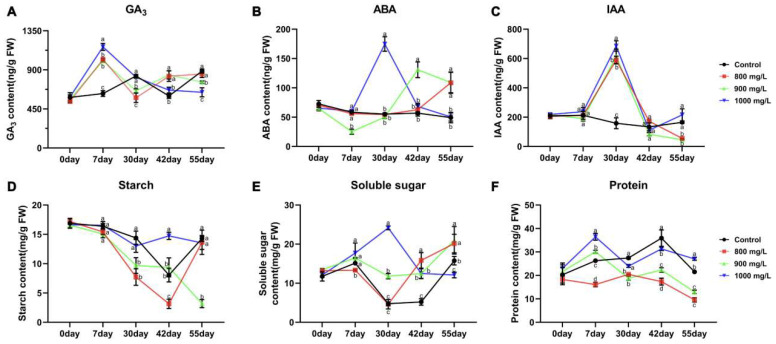
The determination of multiple physiological indicators under different treatments. (**A**) The determination of GA_3_ content. (**B**) The determination of ABA content. (**C**) The determination of IAA content. (**D**) The determination of starch content. (**E**) The determination of soluble sugar content. (**F**) The determination of protein content. Different letters indicated significant differences, Student’s *t* test, *p* < 0.05, n = 3.

**Figure 3 plants-14-01002-f003:**
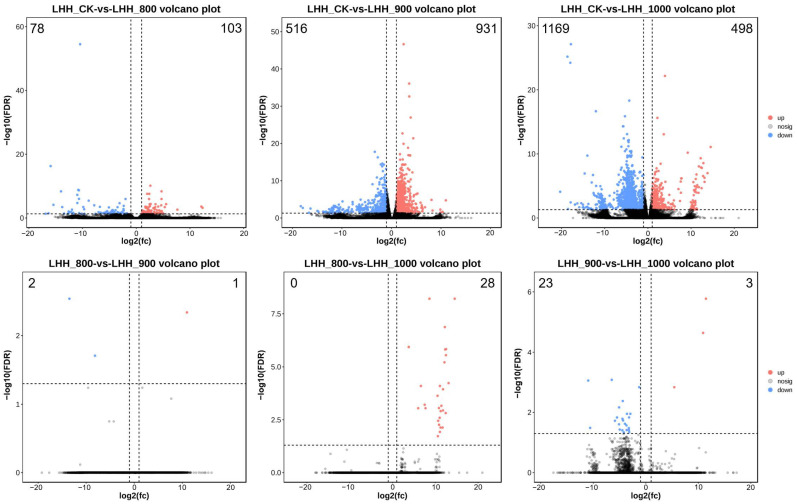
Comparative analysis of DEGs among different treatments. The expression trends of DEGs in the six comparisons. Green circles: down-regulated genes; red circles: up-regulated genes; blue circles: no differential expressed genes. FDR < 0.05, *p* < 0.05 and |fold change| ≥ 2.

**Figure 4 plants-14-01002-f004:**
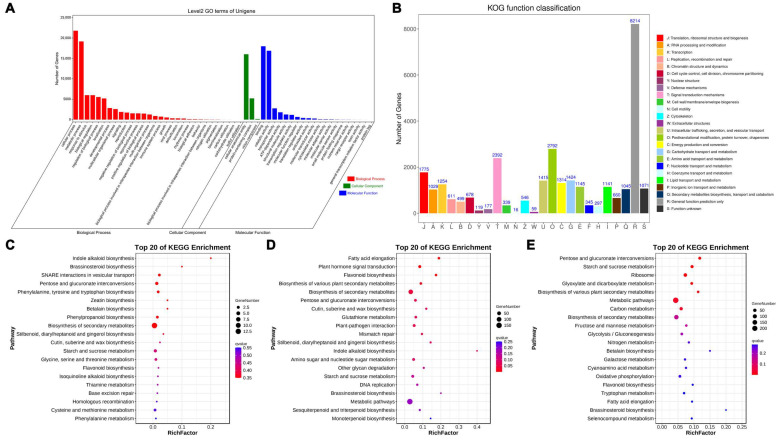
Enrichment analysis of differentially expressed genes among different treatments. (**A**) GO function classification of Unigenes among different treatments. (**B**) KOG function classification of Unigenes among different treatments. (**C**) The top 20 of KEGG pathway enrichments of DEGs in CK vs. 800 mg/L comparison. (**D**) The top 20 of KEGG pathway enrichments of DEGs in CK vs. 900 mg/L comparison. (**E**) The top 20 of KEGG pathway enrichments of DEGs in CK vs. 1000 mg/L comparison.

**Figure 5 plants-14-01002-f005:**
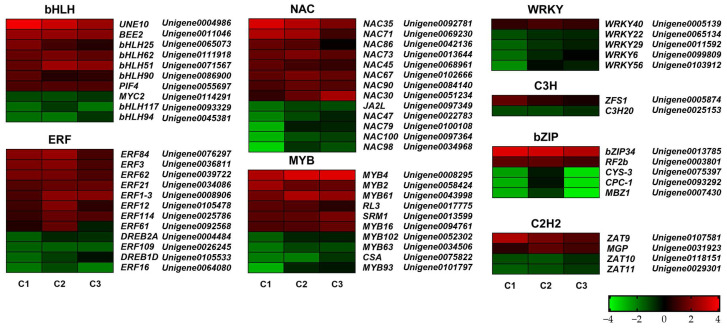
Comparative analysis of DEGs among different exogenous hormone treatments. Red meant the up-regulating and green meant the down-regulating. C1, C2, and C3 represented CK vs. 800 mg/L, CK vs. 900 mg/L and CK vs. 1000 mg/L comparison, respectively. FDR < 0.05, *p* < 0.05 and |fold change| ≥ 2.

**Figure 6 plants-14-01002-f006:**
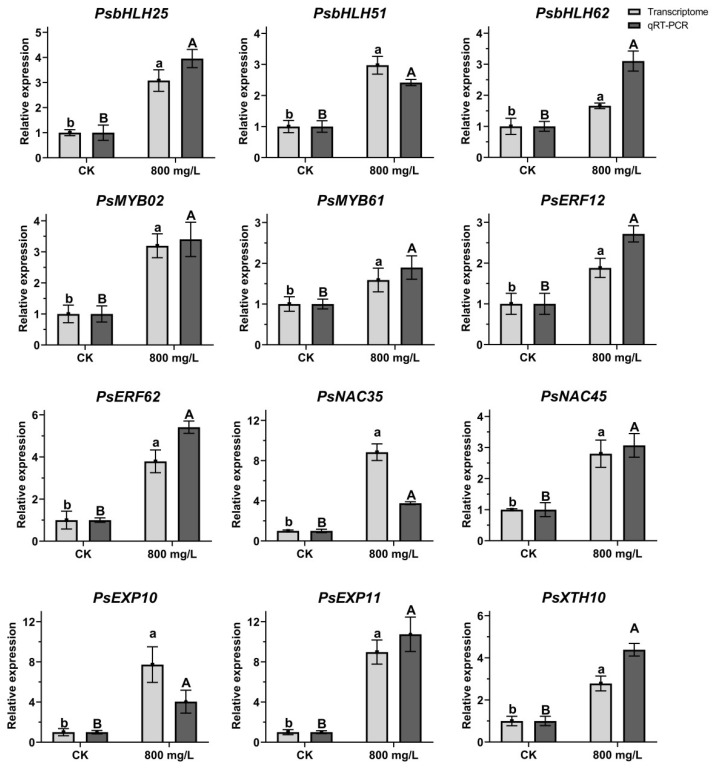
Expression pattern analysis of 12 DEGs. Light grey bars were from the results of transcriptome data; dark grey bars were from the results of qRT-PCR. Different letters indicated significant differences. Means ± SD, n = 3, *p* < 0.05.

**Figure 7 plants-14-01002-f007:**
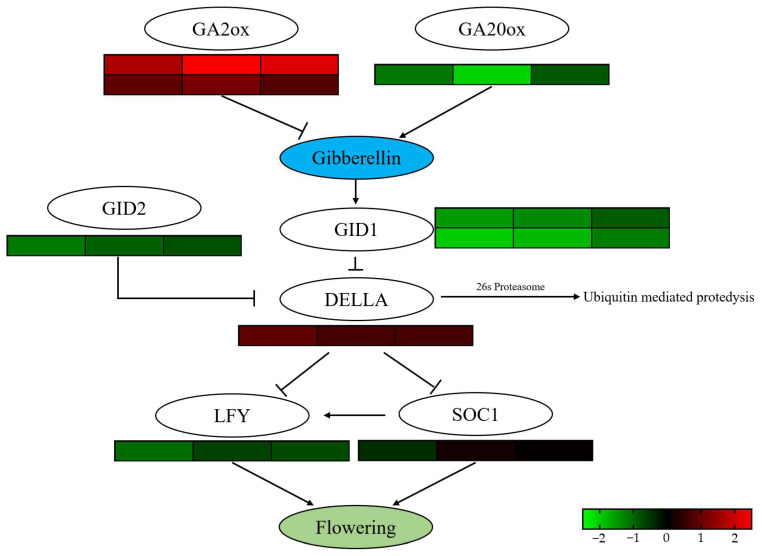
Comparative analysis of DEGs involved in gibberellin signaling pathways under exogenous hormone treatments. The heatmap of plant hormone signal transduction-related DEGs under different treatments. Red meant the up-regulating and green meant the down-regulating. FDR < 0.05, *p* < 0.05 and |fold change| ≥ 2.

**Figure 8 plants-14-01002-f008:**
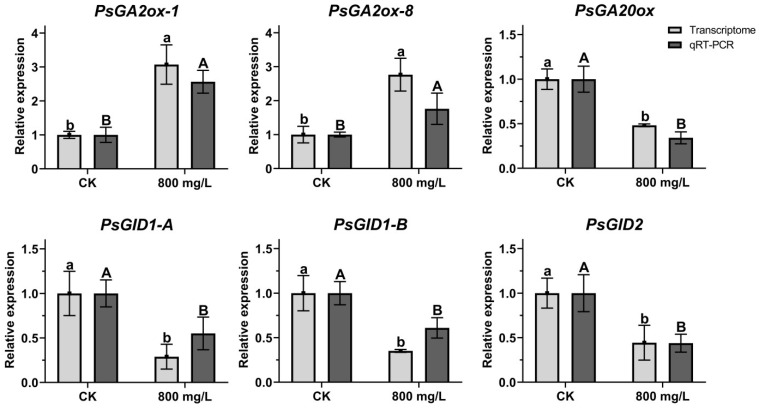
Analysis of gene expression related to GA pathway. Light grey bars were from the results of transcriptome data; dark grey bars were from the results of qRT-PCR. Different letters indicated significant differences. Means ± SD, n = 3, *p* < 0.05.

**Table 1 plants-14-01002-t001:** Phenotypic Observation of Flower Buds. As Treatment 3 and the Control Group did not reach the small bell-shaped bud stage, data for this stage represent flower bud diameter measured at 30 days. Different letters indicated significant differences by vertical comparison, Student’s *t* test, *p* < 0.05, n = 3.

Phenotypic Observation of Flower Buds				
Treatment	Date of Floral Bud Initiation Floral Bud Diameter (cm)	Floral Bud Development Period Floral Bud Diameter (cm)	Bell-Shaped Bud Stage Bell-Shaped Bud Diameter (cm)	First Flowering Date Flower Diameter (cm)
Treatment 1	13 d, 0.83 ± 0.06 a	22 d, 0.97 ± 0.03 a	30 d, 3.40 ± 0.03 a	42 d, 15.47 ± 0.682 a
Treatment 2	13 d, 0.90 ± 0.00 a	22 d, 1.07 ± 0.06 a	33 d, 2.80 ± 0.26 b	45 d, 14.65 ± 0.679 a
Treatment 3	13 d, 0.90 ± 0.00 a	22 d, 0.90 ± 0.02 a	−1.82 ± 0.08 c	-
CK	13 d, 0.87 ± 0.06 a	30 d, 0.87 ± 0.06 a	−1.50 ± 0.20 c	-

**Table 2 plants-14-01002-t002:** Multiple growth parameters in tree peony under different treatments. Different letters indicated significant differences by vertical comparison, Student’s *t* test, *p* < 0.05, n = 3.

Tree Peony Growth Parameters						
	Vegetative Growth			Reproductive Growth		
Treatment	Leaf Width(cm)	Number of Compound Leaves	New Shoot Length (cm)	Total Number of Flowers	Flowering Rate	Flower Diameter (cm)
Treatment 1	4.99 ± 0.735 a	210.21 ± 71.012 ab	11.74 ± 2.063 a	16	0.137 ± 0.152 a	15.47 ± 0.682 a
Treatment 2	4.6 ± 0.893 ab	287.14 ± 87.441 a	9.38 ± 2.704 ab	22	0.186 ± 0.128 a	14.65 ± 0.679 a
Treatment 3	4.05 ± 0.512 bc	290.14 ± 114.988 a	9.81 ± 2.299 ab	2	0.017 ± 0.043 b	-
CK	3.49 ± 0.551 c	181.07 ± 74.523 b	7.66 ± 1.918 b	1	0.01 ± 0.037 b	-

## Data Availability

The data supporting the findings of this study are available from the corresponding author upon request. The data presented in the study are deposited in online repositories. The names of the repository/repositories and accession number(s) can be found below: https://www.ncbi.nlm.nih.gov/, PRJNA1157769 (accessed on 13 March 2025).

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
