# Peer review of "Transcriptomic Analysis of Gibberellin-Mediated Flower Opening Process in Tree Peony (Paeonia suffruticosa)"

_plants, 2025, doi:10.3390/plants14071002_

Round 1
Reviewer 1 Report
Comments and Suggestions for Authors
Dear authors,
1. This statement should be revised “Our findings suggested that exogenous GA3 treatment could accelerate the lower opening process in tree peony through multiple pathways” (L25-26) since only at the appropriate concentration of 800 mg/L and 900 mg/L GA3 showed accelerated lowering while higher than these concentrations, 1000 mg/L GA3 aborted this process (1000 mg/L GA3 treatment had a negative impact on the lower opening). This statement could be represented: "Our findings suggested that an appropriate concentration of exogenous GA3 treatment could accelerate the lower opening process in tree peony through multiple pathways”.
2. Please specify exactly which type of samples (flower bud samples or mature functional leaves located near the terminal buds or other types) were used for the respective analyses as described in the Materials and Methods section. These include in several subsections:
- 2.3. Physiological Measurements
- 2.4. RNA Extraction, Library Construction and Sequencing
- 2.6. RNA Extraction for Quantitative Real-Time PCR Analysis
3. Please provide information on ELISA kits for endogenous hormone quantification (L107) and soluble protein, soluble sugar, and starch content determination (L113-114) in the text. This information includes the product code, product catalog number… etc of Kits. Additionally, similar to other kits mentioned in the section Materials and Methods: Trizol reagent kit (L118-119), RNAprep Pure Plant Kit (L148), and PrimeScriptTM RT reagent Kit (L150).
4. Citation of References should be following the journal’s format. Citing by number instead of authors’ name and reference numbers should be placed in square brackets [ ]. E.g. the citation [Rina and Xiao 2022; Shi 37 et al. 2021] should be reformated as [1, 2].
5. Please provide an access link to the “genome website” (L153).
6. Double-check the qRT-PCR reaction procedure in L159-160: “The reaction procedure was 95 °C for 30s, 95 °C for 5 s, 60 °C for 30 s, a total of 40 cycles; 95 °C for 5 s, 60 °C for 1min, 95 °C for 15s”.
7. Table 2, double-check the column termed "Leaf area (cm)" in this table. Was there the "Leaf area" (cm) or "leaf width" (the widest distance across the leaf blade) as the authors presented in Line 100 of subsection “2.2. Morphological Measurements”?
8. Figures 1 and 4 are low-quality images. Please replace them with high-resolution ones.
9. Why did the authors choose “C2H2, ERF, bHLH, MYB, bZIP, NAC, WRKY, and C3H families” to analyze DEGs in tree peony under different treatments?
Other Remarks
10. Please consider representing the manuscript following the journal’s structure (https://www.mdpi.com/journal/plants/instructions): 1. Introduction -> 2. Results -> 3. Discussion -> 4. Materials and Methods -> 5. Conclusions.
11. Section “Reference” should be presented according to the journal’s style and include an abbreviated journal name, year, volume, and page range as journal format:
“Author 1, A.B.; Author 2, C.D. Title of the article. Abbreviated Journal Name Year, Volume, page range.”
For example, Ref #4:
“Çelikel G F, Doorn V G W (2012) Endogenous ethylene does not regulate opening of unstressed Iris flowers but strongly inhibits it in water-stressed flowers. Journal of Plant Physiology 169(14):1425-1429”
-> “Çelikel, G.F.; Doorn, V.G.W. Endogenous ethylene does not regulate opening of unstressed Iris flowers but strongly inhibits it in water-stressed flowers. J. Plant Physiol. 2012, 169, 1425‒1429”
- Unclear meaning: “holding solution” (L59)
Best regards,

Author Response
Comments 1:This statement should be revised “Our findings suggested that exogenous GA3 treatment could accelerate the lower opening process in tree peony through multiple pathways” (L25-26) since only at the appropriate concentration of 800 mg/L and 900 mg/L GA3 showed accelerated lowering while higher than these concentrations, 1000 mg/L GA3 aborted this process (1000 mg/L GA3 treatment had a negative impact on the lower opening). This statement could be represented: "Our findings suggested that an appropriate concentration of exogenous GA3 treatment could accelerate the lower opening process in tree peony through multiple pathways”.
Response 1:We have revised the corresponding sentences in the article according to the comments.
Comments 2:Please specify exactly which type of samples (flower bud samples or mature functional leaves located near the terminal buds or other types) were used for the respective analyses as described in the Materials and Methods section. These include in several subsections:
- 2.3. Physiological Measurements
- 2.4. RNA Extraction, Library Construction and Sequencing
- 2.6. RNA Extraction for Quantitative Real-Time PCR Analysis
Response 2:Due to the fact that the buds of peonies are mostly mixed buds, the material we used for subsequent experiments was flower buds, which included several experiments such as 2.3, 2.4, and 2.6.
Comments 3:Please provide information on ELISA kits for endogenous hormone quantification (L107) and soluble protein, soluble sugar, and starch content determination (L113-114) in the text. This information includes the product code, product catalog number… etc of Kits. Additionally, similar to other kits mentioned in the section Materials and Methods: Trizol reagent kit (L118-119), RNAprep Pure Plant Kit (L148), and PrimeScriptTM RT reagent Kit (L150).
Response 3:We have added more detailed information about the reagent kit in the article. Please refer to the latest version of the manuscript.
Comments 4:Citation of References should be following the journal’s format. Citing by number instead of authors’ name and reference numbers should be placed in square brackets [ ]. E.g. the citation [Rina and Xiao 2022; Shi 37 et al. 2021] should be reformated as [1, 2].
Response 4:We have revised the reference format according to the journal's requirements. Please refer to the latest version of the manuscript.
Comments 5:Please provide an access link to the “genome website” (L153).
Response 5:Here is the website link (https://ftp.cngb.org/pub/
CNSA/data5/CNP0003098/CNS0560369/CNA0050666), which we have also added to the latest version of the manuscript.
Comments 6:Double-check the qRT-PCR reaction procedure in L159-160: “The reaction procedure was 95 °C for 30s, 95 °C for 5 s, 60 °C for 30 s, a total of 40 cycles; 95 °C for 5 s, 60 °C for 1min, 95 °C for 15s”.
Response 6:We have confirmed that the qRT-PCR program mentioned in the article is indeed correct.
Comments 7:Table 2, double-check the column termed "Leaf area (cm)" in this table. Was there the "Leaf area" (cm) or "leaf width" (the widest distance across the leaf blade) as the authors presented in Line 100 of subsection “2.2. Morphological Measurements”?
Response 7:We have modified the 'leaf width' in Table 2 to be consistent with the one mentioned in the Materials and Methods section.
Comments 8:Figures 1 and 4 are low-quality images. Please replace them with high-resolution ones.
Response 8:We have replaced the relevant images, which should now be clearer and more visible.
Comments 9:Why did the authors choose “C2H2, ERF, bHLH, MYB, bZIP, NAC, WRKY, and C3H families” to analyze DEGs in tree peony under different treatments?
Response 9:Because we have read a large amount of literature and found that genes from these gene families may play a key role in the process of flower opening.
Comments 10:Please consider representing the manuscript following the journal’s structure (https://www.mdpi.com/journal/plants/instructions): 1. Introduction -> 2. Results -> 3. Discussion -> 4. Materials and Methods -> 5. Conclusions.
Response 10:We have revised the format as requested, please refer to the latest version of the manuscript.
Comments 11:Section “Reference” should be presented according to the journal’s style and include an abbreviated journal name, year, volume, and page range as journal format.
Response 11:We have revised the format as requested, please refer to the latest version of the manuscript.
Comments 12:Unclear meaning: “holding solution” (L59)
Response 12:“Holding solution (vase solution)” to provide necessary carbohydrates for fresh cut flowers, in order to prolong their lifespan and ensure continued flowering
Reviewer 2 Report
Comments and Suggestions for Authors
Li et al.,
Dear authors, thank you for the opportunity to read your study. Here, a comprehensible study involving RNA-Seq was developed to study the role of GA3 in the flowering opening of Paeonia suffruticosa. Some minor comments are cited below. In general, the enrichment analysis should be performed only with DEGs (fold-change >2 and <-2, p-value and FDR <0.05), the RNA-Seq should also be validated using correlation analysis, library per library; discussion should be organized in subtopics; and conclusion section should be more focused on insights from RNA-Seq data. Please, provide the suppl. materials for the peer review process.
Recommendation: major corrections are recommended.
Some other comments:
- Abstract: Was the exogenous GA3 application performed in adult plants? under greenhouse or field conditions? Were transcriptome sequencing analyses performed from flower buds or dormant axillary buds? Here, “significantly expressed” changes to “differentially expressed”. Overall, some information related to plant stage, plant growth, plant cultivar, time of GA3 application, tissue sampling, and RNA-Seq should be improved in this topic.
- Introduction: provide genus and family of Paeonia suffruticosa; some citations to be improved or corrected; provide a more detailed and scientific information for “under forcing cultivation”; in addition to “different GA3 treatments” provide information about time of GA3 application; what treatments were performed and evaluated in “transcriptome sequencing was employed”? what is the difference between GA and GA3, for this study? Overall, the last paragraph should be more detailed and clarified;
- Materials and Methods: what is “normal maintenance”? Was GA3 application at 2pm or 800, 900, 1000 mg/L? this is confusing here; when were applied 800, 900, 1000 mg/L? n=3, the n should be in italics; were both flower buds and leaves evaluated here (mature functional leaves (3-6 per group)) by molecular analysis? The method used to determinate the soluble protein content should be more explained; provide detailed information about kits used to soluble protein, soluble sugar, and starch content determination; Why was selected the day 7 for RNA-Seq among other time points? Detailed information about raw data generated for each library should be provided as suppl. table; provide materials and methods of filtering, quality control, and analysis of raw data generated for each library; here, “two different groups” which groups? And here, “between two samples”, which samples? Why? In addition to FDR, p-value <0.05 should also be used; only “fold change ≥ 2” or also <-2? What reference genome was used for RNA-Seq analysis? The first mention of GO should be Gene Ontology (GO); Methods used in the GO analysis should be clarified; This sentence “Genes usually interact with each other to play roles in certain biological functions. Pathway-based analysis helps to further understand genes biological functions. KEGG is the major public pathway-related database (Kanehisa and Goto 2000).” is not methods, then, delete and re-write. Provide detailed information about KEGG enrichment analysis performed here; The same samples used in RNA-Seq were used to RNA isolation and RT-PCR? Provide a reference to genome reference used to retrieve gene sequences; provide a scientific reference to PsACT reference gene; provide a reference to 2−△△CT method; subsection 2.6 lacks information about experimental design (treatments and biological replicates), target genes evaluated, and statistical support for RNA-Seq versus RT-PCR validation, library per library; therefore, it is highly advisable to obtain Transcripts Per Million (TPM) or Fragments Per Kilobase Million (FPKM) values from RNA-Seq, which represent normalized counts akin to qRT-PCR, and correlate them with qRT-PCR values for each gene separately. A noteworthy example of this approach can be found in Table S3 of the referenced article: [https://doi.org/10.3389/fpls.2023.1194244].
- Results: Here, “Four representative stages were identified:” identified for what? Figure 1A should be a diagram showing the experimental design to facilitate understanding of how the study was carried out from the plant to RNA-Seq; Where was used 2ppm of GA3 in Fig. 1? Table 1, title versus columns are confusing; letters of statistical support within Fig. 2 should be increased; Here, “A total of 104,888 differentially expressed genes (DEGs) were identified”, were 104,888 DEGs based on fold-change >2 and <-2, p-value and FDR <0.05? This information is very confusing; Fig. 3 does not show the exact number of DEGs in each contrast; the title of each graph is very confusing; both information should be improved; the legend of Fig. 3 should provide the cut-off for fold-change, p-value, and FDR values; GO and KEGG enrichment analysis, results presented in subsection 3.5, were performed from 104,888 DEGs? The GO, KOG, and KEGG enrichment analysis should be performed only with DEGs identified between each contrast (CK versus 800, CK versus 900, and CK versus 1000) considering fold-change >2 and <-2, and p-value and FDR <0.05; Here, “In the KOG functional classification, 30,357 DEGs were categorized into 25 groups” 30,357 DEGs from what treatment? Figure 4 A and B, DEGs among different treatments, but these two figures do not shown per treatment; please, to be rigorously corrected or deleted; figure 5, provides statistical support within figure (such as p-value and FDR)and, also, gene ID for each gene name; why only 8 transcription factor families were showed? C1, C2, and C3 within Figure 5 and the text of subsection 3.6 should be changed by the full name of each contrast; Fig. 7, provides statistical support (such as p-value and FDR); the RNA-Seq datasets should be deposited in a repository and the number of deposit should be provided here;
- Discussion: before discussion of transcription factors as DEGs, the authors should discuss the different enriched pathways in each contrast, Figure 4 C to 4E, from the top 20, at least the top more representative 5-10 pathways should be dissected, addressing each one in a particular manner; therefore, the discussion section should be organized in subtopics and addressing in each topic at least the GA3, ABA, IAA, 5-10 top enriched pathways, major transcription factors, and major genes involved in flowering opening; in the current version, this topic is poor and enriched pathways are disregarded; citation to be improved or corrected; Here, “however, the expression patterns of related genes differed. This discrepancy could be attributed to differences in the timing of transcriptome sequencing. Previous studies often analyzed samples at 3 DAT, while we sequenced samples at 7 DAT.” provides a more convenient discussion; the conclusion paragraph, last paragraph, should be moved to topic 5 or deleted;
- Conclusion: delete “This study aimed to investigate the role of exogenous GA3 in promoting flower opening in tree peony after dormancy release.”; here, “exogenous GA3 application significantly accelerates” provides concentration and time of application of GA3; the first part of the conclusions are not news, the powerful news are in the transcriptome data; insights from DEGs/transcriptome are poorly presented here (pathways, transcription factors, major genes, others);
- The suppl. material should be provided to the reviewer process; It is important to mention that suppl. material was not provided to the peer review process;
Comments on the Quality of English Language
No comments.
Author Response
Comments 1:
Abstract: Was the exogenous GA3 application performed in adult plants? under greenhouse or field conditions? Were transcriptome sequencing analyses performed from flower buds or dormant axillary buds? Here, “significantly expressed” changes to “differentially expressed”. Overall, some information related to plant stage, plant growth, plant cultivar, time of GA3 application, tissue sampling, and RNA-Seq should be improved in this topic.
Response 1:
â‘ The peony plants used in this study were five years old.
â‘¡All plants were grown under greenhouse conditions. Details regarding these conditions have been added to the revised manuscript.
â‘¢Transcriptome analysis was conducted on flower buds following dormancy break.
â‘£The Materials and Methods section has been updated to improve the description of material handling procedures.
Comments 2:
Introduction: provide genus and family of Paeonia suffruticosa; some citations to be improved or corrected; provide a more detailed and scientific information for “under forcing cultivation”; in addition to “different GA3 treatments” provide information about time of GA3 application; what treatments were performed and evaluated in “transcriptome sequencing was employed”? what is the difference between GA and GA3, for this study? Overall, the last paragraph should be more detailed and clarified;
Response 2:
â‘ Peony belongs to the family Paeoniaceae and the genus Paeonia.
â‘¡We have revised the relevant references and included details of our greenhouse cultivation conditions in the Materials and Methods section.
â‘£We have revised the information regarding the timing of GA3 application in the Materials and Methods section.
⑤Transcriptome sequencing analysis was performed on all treatments, including CK, 800, 900, and 1000 mg/L.
â‘¥Gibberellins comprise a large group of compounds, and GA3 is among the most biologically active in studies involving peonies.
Comments 3:
Materials and Methods: what is “normal maintenance”? Was GA3 application at 2pm or 800, 900, 1000 mg/L? this is confusing here; when were applied 800, 900, 1000 mg/L? n=3, the n should be in italics; were both flower buds and leaves evaluated here (mature functional leaves (3-6 per group)) by molecular analysis? The method used to determinate the soluble protein content should be more explained; provide detailed information about kits used to soluble protein, soluble sugar, and starch content determination; Why was selected the day 7 for RNA-Seq among other time points? Detailed information about raw data generated for each library should be provided as suppl. table; provide materials and methods of filtering, quality control, and analysis of raw data generated for each library; here, “two different groups” which groups? And here, “between two samples”, which samples? Why? In addition to FDR, p-value <0.05 should also be used; only “fold change ≥ 2” or also <-2? What reference genome was used for RNA-Seq analysis? The first mention of GO should be Gene Ontology (GO); Methods used in the GO analysis should be clarified; This sentence “Genes usually interact with each other to play roles in certain biological functions. Pathway-based analysis helps to further understand genes biological functions. KEGG is the major public pathway-related database (Kanehisa and Goto 2000).” is not methods, then, delete and re-write. Provide detailed information about KEGG enrichment analysis performed here; The same samples used in RNA-Seq were used to RNA isolation and RT-PCR? Provide a reference to genome reference used to retrieve gene sequences; provide a scientific reference to PsACT reference gene; provide a reference to 2−△△CT method; subsection 2.6 lacks information about experimental design (treatments and biological replicates), target genes evaluated, and statistical support for RNA-Seq versus RT-PCR validation, library per library; therefore, it is highly advisable to obtain Transcripts Per Million (TPM) or Fragments Per Kilobase Million (FPKM) values from RNA-Seq, which represent normalized counts akin to qRT-PCR, and correlate them with qRT-PCR values for each gene separately. A noteworthy example of this approach can be found in Table S3 of the referenced article: [https://doi.org/10.3389/fpls.2023.1194244].
Response 3:
â‘ “normal maintenance”:A greenhouse environment with 14 hours of light, 10 hours of darkness and an average temperature of 24 degrees.
â‘¡We applied GA3 treatment at 14:00 PM.
â‘¢We changed the relevant text format.
â‘£We only evaluated flower buds by molecular analysis.
⑤We updated the product codes of all relevant kits in the text, and refer to the instructions attached to the kit for experimental methods.
â‘¥We wanted to screen transcriptional regulators that could rapidly respond to GA signals, so we chose the seventh day as the time of RNA-Seq sequencing.
⑦We have submitted the details of the original data in the appendix, see the supplementary appendix for details.
â‘§"Two different groups" refers to the comparative analysis between CK and three treatment groups to explore the differential genes.
⑨Our main reference indicators are mainly FDR indicators. Fold change we use absolute values, so FC ≥ 2 or FC ≤ -2 are within our scope of application. Similarly, the p-value < 0.05, and we have supplemented these screening conditions.
â‘©Here is the website link (https://ftp.cngb.org/pub/CNSA/data5/CNP0003098/CNS0560369/CNA0050666), which we have also added to the latest version of the manuscript.
⑪We have added the process of go enrichment analysis.
â‘«This sentence has been changed and rewritten according to your modification comments.
⑬RNA-Seq samples and qRT-PCR samples were two samples with three replicates each.
â‘This is our reference in response to this sentence “Provide a reference to genome reference used to retrieve gene sequences; provide a scientific reference to PsACT reference gene; provide a reference to 2−△△CT method.”(https://doi.org/10.3390/plants13030437)
â‘®In Figure 6 and figure 8, we compared the qRT-PCR value and transcriptome value of genes worthy of attention, so we thought that we might not need to map additionally.
Comments 4:
Results: Here, “Four representative stages were identified:” identified for what? Figure 1A should be a diagram showing the experimental design to facilitate understanding of how the study was carried out from the plant to RNA-Seq; Where was used 2ppm of GA3 in Fig. 1? Table 1, title versus columns are confusing; letters of statistical support within Fig. 2 should be increased; Here, “A total of 104,888 differentially expressed genes (DEGs) were identified”, were 104,888 DEGs based on fold-change >2 and <-2, p-value and FDR <0.05? This information is very confusing; Fig. 3 does not show the exact number of DEGs in each contrast; the title of each graph is very confusing; both information should be improved; the legend of Fig. 3 should provide the cut-off for fold-change, p-value, and FDR values; GO and KEGG enrichment analysis, results presented in subsection 3.5, were performed from 104,888 DEGs? The GO, KOG, and KEGG enrichment analysis should be performed only with DEGs identified between each contrast (CK versus 800, CK versus 900, and CK versus 1000) considering fold-change >2 and <-2, and p-value and FDR <0.05; Here, “In the KOG functional classification, 30,357 DEGs were categorized into 25 groups” 30,357 DEGs from what treatment? Figure 4 A and B, DEGs among different treatments, but these two figures do not shown per treatment; please, to be rigorously corrected or deleted; figure 5, provides statistical support within figure (such as p-value and FDR)and, also, gene ID for each gene name; why only 8 transcription factor families were showed? C1, C2, and C3 within Figure 5 and the text of subsection 3.6 should be changed by the full name of each contrast; Fig. 7, provides statistical support (such as p-value and FDR); the RNA-Seq datasets should be deposited in a repository and the number of deposit should be provided here;
Response 4:
â‘ There are multiple developmental stages in the flowering process. We selected these four stages with significant phenotypic changes for observation.
â‘¡We have changed the title of Table 1 to “Phenotypic Observation of Flower Buds”.
â‘¢We mainly observed the changes of endogenous hormone content under different treatments at the same developmental stage, so we only marked the differences between different treatments.
â‘£We additionally provided supplementary materials, see supplementary materials for details.
⑤In the volcano diagram in Figure 3, we additionally marked the number of up-regulated genes and down regulated genes.
â‘¥We added the corresponding values in the legend of Figure 3. KEGG analysis was carried out from 104888 unigenes. Our comparison is indeed the result of the comparison between CK and the other three treatment groups. The number of 30357 should be the number of Unigene, which we have corrected.
⑦We modified the legend to make the expression of figure 4A and B clearer.
â‘§In the legend of Figure 5, we have additionally provided p-values, fold change values, and annotated the gene names.
⑨Because we believe that the genes in these eight gene families may be closely related to the flower opening process.。
â‘©We have explained what C1, C2, and C3 represent in the legend of Figure 5. We believe using abbreviations makes the figure more concise and clear.
⑪We have additionally provided p-values and fold change values in the legend of Figure 7.
â‘«We have provided the repository link for the RNA-Seq data in the updated " Data Availability " section.
Comments 5:
Discussion: before discussion of transcription factors as DEGs, the authors should discuss the different enriched pathways in each contrast, Figure 4 C to 4E, from the top 20, at least the top more representative 5-10 pathways should be dissected, addressing each one in a particular manner; therefore, the discussion section should be organized in subtopics and addressing in each topic at least the GA3, ABA, IAA, 5-10 top enriched pathways, major transcription factors, and major genes involved in flowering opening; in the current version, this topic is poor and enriched pathways are disregarded; citation to be improved or corrected; Here, “however, the expression patterns of related genes differed. This discrepancy could be attributed to differences in the timing of transcriptome sequencing. Previous studies often analyzed samples at 3 DAT, while we sequenced samples at 7 DAT.” provides a more convenient discussion; the conclusion paragraph, last paragraph, should be moved to topic 5 or deleted;
Response 5:
â‘ We have added a discussion of the enrichment analysis to the Discussion section, and we have added and revised some references.
â‘¡We have revised the wording of this sentence. “however, the expression patterns of related genes differed. This discrepancy could be attributed to differences in the timing of transcriptome sequencing. Previous studies often analyzed samples at 3 DAT, while we sequenced samples at 7 DAT.”
â‘¢We have deleted the last paragraph.
Comments 6:
Conclusion: delete “This study aimed to investigate the role of exogenous GA3 in promoting flower opening in tree peony after dormancy release.”; here, “exogenous GA3 application significantly accelerates” provides concentration and time of application of GA3; the first part of the conclusions are not news, the powerful news are in the transcriptome data; insights from DEGs/transcriptome are poorly presented here (pathways, transcription factors, major genes, others);
Response 6:
â‘ We have deleted this sentence. “This study aimed to investigate the role of exogenous GA3 in promoting flower opening in tree peony after dormancy release.”
â‘¡The treatment methods and timing are described in the Materials and Methods section; therefore, we have not mentioned them again here.
â‘¢We have revised the content of the Conclusion section.
Comments 7:
The suppl. material should be provided to the reviewer process; It is important to mention that suppl. material was not provided to the peer review process;
Response 7:
We have provided supplementary files. Please refer to the latest submission.
Round 2
Reviewer 1 Report
Comments and Suggestions for Authors
Dear authors:
Many thanks for the revisions to your manuscript. The authors have efforts to improve the manuscript. The revised version has been greatly improved. I have only one remark for the revised version before publication.
Remark:
- Section “Reference” should be presented according to the journal’s style with abbreviated journal name, year, volume, and page range as journal format:
“Author 1, A.B.; Author 2, C.D. Title of the article. Abbreviated Journal Name Year, Volume, page range.”
For example, Ref #4:
“Çelikel G F, Doorn V G W (2012) Endogenous ethylene does not regulate opening of unstressed Iris flowers but strongly inhibits it in water-stressed flowers. Journal of Plant Physiology 169(14):1425-1429”
-> “Çelikel, G.F.; Doorn, V.G.W. Endogenous ethylene does not regulate opening of unstressed Iris flowers but strongly inhibits it in water-stressed flowers. J. Plant Physiol. 2012, 169, 1425‒1429”
- My bad in Round 1. Please revise “lower” (Line 27) to “flower”.
Good luck and Best regards,

Reviewer 2 Report
Comments and Suggestions for Authors
Dear authors, thank you for providing the improved version of this manuscript. The authors have satisfactorily addressed and improved this manuscript. Therefore. this reviewer consider this study appropriated or suitable for publication in the PLANTS.